# AI-Driven Prediction of Glasgow Coma Scale Outcomes in Anterior Communicating Artery Aneurysms

**DOI:** 10.3390/jcm14082672

**Published:** 2025-04-14

**Authors:** Corneliu Toader, Octavian Munteanu, Mugurel Petrinel Radoi, Carla Crivoi, Razvan-Adrian Covache-Busuioc, Matei Serban, Alexandru Vlad Ciurea, Nicolaie Dobrin

**Affiliations:** 1Department of Neurosurgery, “Carol Davila” University of Medicine and Pharmacy, 020021 Bucharest, Romania; corneliu.toader@umfcd.ro (C.T.); razvan-adrian.covache-busuioc0720@stud.umfcd.ro (R.-A.C.-B.); matei.serban2021@stud.umfcd.ro (M.S.); prof.avciurea@gmail.com (A.V.C.); 2Department of Vascular Neurosurgery, National Institute of Neurology and Neurovascular Diseases, 077160 Bucharest, Romania; 3Department of Anatomy, “Carol Davila” University of Medicine and Pharmacy, 050474 Bucharest, Romania; 4Department of Computer Science, Faculty of Mathematics and Computer Science, University of Bucharest, 010014 Bucharest, Romania; crivoicarla02@gmail.com; 5Puls Med Association, 051885 Bucharest, Romania; 6Neurosurgery Department, Sanador Clinical Hospital, 010991 Bucharest, Romania; 7Medical Section, Romanian Academy, 010071 Bucharest, Romania; 8“Nicolae Oblu” Clinical Hospital, 700309 Iasi, Romania; dobrin_nicolaie@yahoo.com

**Keywords:** artificial intelligence, machine learning in healthcare, Glasgow Coma Scale prediction, explainable AI, random forest, neurological outcome prediction, precision medicine, clinical decision support systems, critical care analytics

## Abstract

**Background**: The Glasgow Coma Scale (GCS) is a cornerstone in neurological assessment, providing critical insights into consciousness levels in patients with traumatic brain injuries and other neurological conditions. Despite its clinical importance, traditional methods for predicting GCS scores often fail to capture the complex, multi-dimensional nature of patient data. This study aims to address this gap by leveraging machine learning (ML) techniques to develop accurate, interpretable models for GCS prediction, enhancing decision making in critical care. **Methods**: A comprehensive dataset of 759 patients, encompassing 25 features spanning pre-, intra-, and post-operative stages, was used to develop predictive models. The dataset included key variables such as cognitive impairments, Hunt and Hess scores, and aneurysm dimensions. Six ML algorithms, including random forest (RF), XGBoost, and artificial neural networks (ANN), were trained and rigorously evaluated. Data preprocessing involved numerical encoding, standardization, and stratified splitting into training and validation subsets. Model performance was assessed using accuracy and receiver operating characteristic area under the curve (ROC AUC) metrics. **Results**: The RF model achieved the highest accuracy (86.4%) and mean ROC AUC (0.9592 ± 0.0386, standard deviation), highlighting its robustness and reliability in handling heterogeneous clinical datasets. XGBoost and SVM models also demonstrated strong performance (ROC AUC = 0.9502 and 0.9462, respectively). Key predictors identified included the Hunt and Hess score, aneurysm dimensions, and post-operative factors such as prolonged intubation. Ensemble methods outperformed simpler models, such as K-nearest neighbors (KNN), which struggled with high-dimensional data. **Conclusions**: This study demonstrates the transformative potential of ML in GCS prediction, offering accurate and interpretable tools that go beyond traditional methods. By integrating advanced algorithms with clinically relevant features, this work provides a dynamic, data-driven framework for critical care decision making. The findings lay the groundwork for future advancements, including multi-modal data integration and broader validation, positioning ML as a vital tool in personalized neurological care.

## 1. Introduction

The Glasgow Coma Scale (GCS) has remained an indispensable tool in clinical neurology for nearly five decades. It provides a standardized framework for assessing the level of consciousness in patients with traumatic brain injuries, intracranial aneurysms, or other neurological conditions [1]. Despite its simplicity and ease of use, the GCS’s interpretation often relies on subjective assessments and static scoring, which may fall short in capturing the dynamic and multifaceted nature of a patient’s neurological status. The GCS evaluates three key neurological functions: eye opening (E), verbal response (V), and motor response (M). Each component is scored separately—eye opening from 1 (no response) to 4 (spontaneous), verbal response from 1 (no verbal output) to 5 (oriented conversation), and motor response from 1 (no movement) to 6 (obeys commands). These are then summed to yield a total GCS score ranging from 3 (deep unconsciousness) to 15 (fully alert and oriented). Lower scores are indicative of more severe neurological impairment and are associated with worse clinical outcomes. The scale is widely used across emergency and critical care settings to track neurological status, stratify patient severity, and support treatment decisions. Despite its simplicity and ease of use, the GCS’s interpretation often relies on subjective assessments and static scoring, which may fall short in capturing the dynamic and multifaceted nature of a patient’s neurological status [2]. This limitation becomes particularly evident in critical care settings, where timely and accurate predictions of GCS scores could guide interventions, improve resource allocation, and inform long-term prognoses [3].

Predicting GCS scores is more than just a numerical exercise—it is a vital step in advancing precision medicine. A low GCS score signals critical deterioration, necessitating immediate medical intervention, while higher scores may predict better recovery trajectories [4]. However, traditional prediction methods, which often rely on linear statistical models or expert-driven heuristics, fail to fully account for the inherent complexity of clinical data [5]. Neurological outcomes are influenced by a wide array of factors, including demographic attributes, pre-existing comorbidities, surgical metrics, and post-operative complications. Capturing the intricate interactions among these variables requires sophisticated computational approaches that can go beyond traditional paradigms [6].

Machine learning (ML), a subset of artificial intelligence, has emerged as a transformative force in healthcare. By leveraging algorithms capable of learning from data, ML models can uncover non-linear relationships, interactions, and patterns that may elude conventional statistical methods [7,8]. In recent years, ML has demonstrated exceptional promise in predicting patient outcomes across diverse domains, from early detection of sepsis and cardiac arrest to assessing functional recovery in stroke patients. However, the application of ML to predicting GCS scores specifically has been underexplored. This presents a significant opportunity for innovation, particularly as advanced computational techniques are now more accessible than ever [9].

Recent advancements in ensemble methods such as Random Forest (RF) and XGBoost, as well as deep learning architectures like Artificial Neural Networks (ANNs), have paved the way for highly accurate and interpretable predictions [10]. Ensemble methods aggregate the outputs of multiple models to reduce variance and bias, making them particularly robust for clinical datasets with mixed feature types (e.g., categorical and continuous variables) [11]. On the other hand, neural networks excel at capturing intricate, non-linear relationships, although they often face challenges with interpretability and overfitting in smaller datasets. These complementary strengths provide a strong foundation for tackling the complexities of GCS prediction [12].

Our study is uniquely positioned to address this gap by developing and validating ML models tailored specifically for GCS prediction. The dataset used in this research comprises 759 patients, with a rich array of 25 features spanning pre-surgical conditions (e.g., cognitive impairments, memory deficits), surgical factors (e.g., aneurysm dimensions, Hunt and Hess scores), and post-operative outcomes (e.g., hydrocephalus, prolonged intubation). This comprehensive dataset provides a unique opportunity to integrate domain knowledge with state-of-the-art ML techniques, enabling the creation of models that are not only highly accurate but also clinically interpretable.

A key focus of this work is interpretability—a feature often overlooked in many ML applications in medicine. For a predictive model to gain traction in clinical settings, it must not only outperform traditional methods in accuracy but also provide clear insights into the factors driving its predictions. By using feature importance analysis and other interpretability methods, this study seeks to balance model complexity with clinical usability, making predictions more transparent and accessible for clinicians.

The novelty of this research lies in its targeted application of ML to predict GCS scores, a critical yet understudied area in neurological assessment. Existing studies in the domain of ML in neurology have largely focused on broader outcomes, such as mortality or long-term disability, leaving a significant gap in the precise prediction of consciousness levels. By addressing this gap, our work has the potential to enhance decision making in critical care and neurosurgical contexts, ultimately improving patient outcomes and advancing the field of precision medicine.

This study applies ML to predict GCS outcomes, focusing on improving both accuracy and interpretability in neurological assessment. While ML has been used in neurological prognosis, most models have targeted mortality or long-term functional recovery, rather than real-time consciousness evaluation in acute care. Predicting GCS presents specific challenges, as it requires models to process pre-, intra-, and post-operative clinical variables to generate timely and clinically meaningful insights. To address this, we developed ML models tailored for GCS score prediction, incorporating ensemble learning techniques to enhance performance across different patient groups. Additionally, feature importance analysis provides insights into key clinical predictors, supporting more transparent and interpretable decision making. By refining GCS prediction with a structured, data-driven approach, this study contributes to the advancement of ML-based neurological assessment in critical care.

Statistical models traditionally used in neurological assessment often rely on simplified assumptions, which may not fully account for the variability and complexity of patient outcomes. ML provides an alternative by capturing nonlinear relationships between clinical factors, allowing for a more refined evaluation of neurological status. In particular, ensemble learning and ANNs offer a way to analyze patterns that may not be immediately evident through conventional methods, potentially improving predictive stability across diverse patient populations. A critical aspect of implementing ML in clinical settings is ensuring that predictions remain interpretable and aligned with clinical reasoning. Models that lack transparency may hinder adoption, as clinicians require clear justifications for automated recommendations. By focusing on feature importance analysis, this study provides a structured approach to identifying which variables contribute most to predictive accuracy, making the results more accessible and actionable. While additional validation is necessary, integrating ML-driven insights into routine neurological assessment could enhance early risk identification and personalized patient management strategies.

## 2. Materials and Methods

### 2.1. Study Design and Dataset Description

This study aimed to develop predictive models for the GCS using a clinical dataset of 759 patients, curated to encompass a wide range of relevant medical attributes. The dataset was retrospectively collected from electronic medical records at the National Institute of Neurology and Neurovascular Diseases in Bucharest, Romania. All patients included in the study were admitted with a confirmed diagnosis of AComA aneurysm and underwent surgical intervention at this center. Clinical information was extracted from operative reports, post-operative monitoring data, and discharge summaries by a trained data curation team, under ethical approval (Project No. 6971).

Inclusion criteria comprised adult patients (≥18 years) with complete documentation of pre-operative clinical history, surgical intervention details, and post-operative neurological evaluations, including Glasgow Coma Scale (GCS) scores. Exclusion criteria included patients with missing data, aneurysms in non-AComA locations, unruptured aneurysms, or concurrent neurological conditions (e.g., tumors, infections) that could confound GCS interpretation. All data were anonymized and processed in compliance with institutional privacy protocols and GDPR regulations.

The dataset included 25 clinically relevant variables spanning pre-operative conditions (e.g., obesity, arterial hypertension, cognitive impairments, memory impairments, and visual disturbances), intra-operative metrics (e.g., aneurysm diameter and neck size), and post-operative outcomes (e.g., prolonged intubation, hydrocephalus, or neurological decline). These variables provided a comprehensive representation of the clinical trajectory across the continuum of care, enhancing the quality and richness of predictive modeling.

The dataset is not publicly available due to institutional and privacy constraints; however, it may be shared upon reasonable request and with appropriate ethical approvals. Because the dataset originates from a single neurosurgical institution, we acknowledge that this limits the generalizability of our findings. Institutional practices, referral patterns, and patient demographics may influence model performance. Consequently, while our models demonstrated strong internal validity, external validation across other centers and populations is necessary before widespread clinical implementation. This limitation underscores the need for collaborative, multi-institutional datasets to improve model robustness and cross-population applicability in future work.

To support the development of predictive models, the Glasgow Coma Scale (GCS) scores were stratified into five clinically relevant categories: <5, 5–8, 9–12, 13–14, and 15. This categorical segmentation facilitated more nuanced prognostic assessments and allowed for the application of machine learning algorithms tailored to distinct levels of consciousness. As such, the target variable encompassed a wide range of clinical severities, thereby enhancing the robustness and applicability of the prediction models. A detailed demographic analysis was conducted to evaluate the representativeness of the patient cohort. The study included 759 individuals, with a near-equal gender distribution—400 females (52.7%) and 359 males (47.3%)—and an age span ranging from 20 to over 80 years. The highest concentration of patients was observed in the 50–60-year age group. Age distribution patterns were consistent across sexes, although a modest predominance of older female patients was noted. Similarly, both urban and rural subgroups demonstrated comparable age-related trends, with peak incidence occurring in middle-aged individuals (Table 1).

### 2.2. Data Preprocessing

Data preprocessing was performed to ensure compatibility with ML algorithms and improve interpretability. Continuous variables were discretized into clinically meaningful bins. For instance, aneurysm diameter was categorized as <6 mm, 6–12 mm, 12–20 mm, 20–25 mm, and >25 mm, while aneurysm neck size was grouped into <1 mm, 1–3 mm, 3–5 mm, 5–8 mm, and >8 mm. These categorizations aligned with standard clinical practices, making the dataset more interpretable for both models and clinicians.

Categorical variables were encoded numerically, and all features were standardized to eliminate scale-related biases. This transformation ensured that the algorithms treated each variable consistently, regardless of its original scale or range. These preprocessing steps established a foundation for model training and evaluation. The decision to discretize continuous variables into clinically relevant bins was made to enhance both model interpretability and alignment with established medical guidelines. For aneurysm diameter and neck size, these predefined thresholds reflect well-recognized risk stratifications in neurosurgical practice, where size-based classifications influence treatment strategies, rupture risk assessment, and post-surgical prognosis. By grouping values into structured categories, the models leverage these medical distinctions rather than treating the variables as purely numerical, where minor variations might not hold clinical significance.

Additionally, binning reduces sensitivity to outliers and data sparsity issues, which can impact model performance, particularly in datasets of limited size. Given the heterogeneous nature of the dataset, maintaining clinically meaningful feature representation helps prevent overfitting while preserving the interpretability of the predictions. However, recognizing the trade-off between granularity and predictive power, future refinements could explore hybrid approaches, such as retaining continuous representations in tree-based models while utilizing categorical groupings for interpretability-focused analyses.

### 2.3. Data Splitting

The dataset was divided into two subsets to enable unbiased evaluation of model performance. A stratified sampling approach was employed to allocate 80% of the data (607 patients) to the training set and 20% (152 patients) to the validation set. This method preserved the distribution of key outcomes, such as the Glasgow Outcome Scale (GOS), across both subsets. For example, the training set included 289 cases of Grade 5 outcomes, while the validation set contained 134 cases, ensuring that both subsets accurately represented the clinical spectrum of the data. This meticulous splitting process minimized sampling bias and supported the development of generalizable predictive models. While stratified sampling helped preserve the distribution of key clinical outcomes, the single-center design of this study may limit the generalizability of the findings to broader patient populations. Institutional factors such as variations in surgical protocols, post-operative care, and patient demographics could introduce biases that affect model performance in external settings.

Additionally, although stratification balanced outcome distributions, subtle biases in patient characteristics may still exist. For instance, differences in referral patterns or center-specific treatment approaches could influence the relative proportions of certain subgroups, potentially affecting how well the model generalizes to datasets from other institutions.

### 2.4. Algorithm Description

To explore the predictive potential of the dataset, six ML algorithms were implemented, each selected for its ability to capture distinct patterns and complexities.

The K-nearest neighbors (KNN) algorithm, known for its simplicity, classified patients based on proximity to the most similar cases. We have employed this model as baseline. Support vector machine (SVM) employed a linear kernel to create hyperplanes that separated classes effectively. Ensemble methods, such as extra trees regressor (ETR) and RF, utilized multiple decision trees to capture complex patterns, with RF reducing overfitting through ensemble averaging. The XGBoost algorithm optimized gradient boosting, offering high efficiency and precision for large-scale datasets. Finally, the ANN, a deep learning model with three dense layers (64, 64, and 32 units), utilized ReLU activation functions to capture non-linear relationships. Dropout regularization (25%) was introduced between layers to prevent overfitting, and the model was trained using the Adam optimizer with a categorical cross-entropy loss function.

Each algorithm underwent GridSearch hyperparameter tuning to optimize performance. For instance, the best configuration for KNN included n_neighbors = 8, while SVM utilized a linear kernel. The RF and ETR algorithms performed best with n_estimators = 19 and n_estimators = 16, respectively, and XGBoost was tuned with parameters such as learning_rate = 0.12, colsample_bytree = 0.4, and max_depth = 180. The architecture of the ANN was also fine-tuned, balancing layer size and dropout rates to maximize predictive accuracy.

All models were developed and evaluated using Python 3.12. The computational environment included key libraries such as NumPy (v1.26.4) for numerical processing, Matplotlib (v3.9.0) and Matplotlib-Venn (v0.11.9) for visualization, Scikit-learn (v1.5.0) for traditional machine learning models and evaluation, XGBoost (v2.0.3) for gradient boosting implementation, and TensorFlow (v2.16.1) and TensorFlow-Intel (v2.16.1) for deep learning model development. All code execution was performed in a secure, local environment in compliance with institutional data privacy protocols.

The computations were executed on a local workstation equipped with an Intel^®^ Core™ i7-13620H processor (manufactured by Intel Corporation, Santa Clara, CA, USA, 13th Gen, 2.40 GHz, 64-bit) and 16 GB of RAM. This hardware configuration was sufficient to support all training and evaluation tasks, including those involving deep learning models. No high-performance computing clusters or cloud-based infrastructure were utilized, ensuring that the experiments can be reproduced on standard academic or clinical research machines.

### 2.5. Model Evaluation

The performance of each algorithm was evaluated using accuracy and the receiver operating characteristic area under the curve (ROC AUC) metric, measured on the test dataset. These metrics provided a comprehensive view of model performance, capturing both predictive power and consistency.

In terms of accuracy, the RF algorithm emerged as the best-performing model, achieving 86.4% accuracy on the test dataset. SVM and XGBoost closely followed with accuracies of 85.9%, while the ETR reached 85.1%. The ANN demonstrated strong performance with 83.8% accuracy, though it slightly trailed the deep learning methods. KNN recorded the lowest accuracy at 78.9%, reflecting its limitations in handling the dataset’s complexity.

The ROC AUC scores reinforced these findings, highlighting the models’ ability to distinguish between GCS categories. RF achieved the highest mean ROC AUC (0.9592, SD: 0.0386), followed by ANN (0.9533, SD: 0.0394) and XGBoost (0.9502, SD: 0.0405). These results emphasized the consistency and robustness of deep learning methods for this application. To further evaluate the statistical significance of the differences in model performance, we applied the DeLong test. Since this test is inherently designed for binary classification, we adopted a one-vs-rest strategy for each GCS class across all models [13].

Hyperparameter selection was conducted using grid search with five-fold cross-validation, optimizing model performance while preventing overfitting. Parameters were tuned based on validation accuracy and ROC AUC, ensuring effective generalization. In the ANN, dropout rates of 10%, 25%, and 50% were tested, with 25% chosen as the best balance between regularization and stability. The hidden layer sizes (64, 64, and 32 units) were determined through iterative tuning to maximize predictive performance without excessive complexity. The SVM was evaluated with linear and radial basis function (RBF) kernels, with the linear kernel selected due to its faster convergence and better generalization on this structured dataset. For ensemble models, the number of estimators was adjusted to balance predictive power and efficiency, resulting in n_estimators = 19 for RF and n_estimators = 16 for ETR. XGBoost was optimized with a learning rate of 0.12, colsample_bytree of 0.4, and max_depth of 180, ensuring controlled complexity while enhancing feature utilization.

These optimizations ensured that each model was tailored to the dataset’s characteristics, improving both accuracy and interpretability.

## 3. Results

### 3.1. Dataset Characteristics

The dataset used in this study contained data from 759 patients, with all variables being complete (non-null), ensuring the integrity and reliability of the analysis. The dataset comprised 25 features across various clinical dimensions, including pre-surgical conditions, surgical parameters, and post-surgical outcomes. Among these variables, three were categorical, eight were continuous (float64), and the remaining were integer-based (int64). This diversity of data types required preprocessing and transformation to optimize their use in predictive modeling.

Key pre-surgical features included obesity, arterial hypertension, diabetes, and neurological symptoms such as cognitive impairment, memory impairment, and visual disturbances before surgery. Additionally, more granular details such as hemiparesis, comitial seizures, and aphasia before surgery provided a nuanced view of patients’ pre-operative states. Two significant surgical features, aneurysm diameter and aneurysm neck size, were treated as categorical variables to align with clinical thresholds and facilitate interpretation.

Post-operative outcomes, which included unsystematized balance disorders, memory impairment, and visual disturbances after surgery, were recorded as continuous variables, allowing detailed tracking of recovery or deterioration. Other critical variables, such as the Hunt and Hess scores and the necessity for prolonged intubation after surgery (>96 h), added depth to the dataset, providing insights into the severity and progression of conditions.

This rich collection of variables offered a comprehensive foundation for predictive modeling, ensuring that the ML algorithms could capture the multifaceted nature of clinical outcomes.

### 3.2. Model Accuracy on the Test Dataset

The test dataset evaluation revealed distinct performance patterns across the six ML algorithms, reflecting their ability to leverage the dataset’s diverse features. Among the models, the RF algorithm achieved the highest accuracy at 86.4%, outperforming the other methods. Both SVM and XGBoost followed closely, with accuracies of 85.9%, indicating their strength in handling a combination of categorical, continuous, and integer-based data.

The ETR achieved an accuracy of 85.1%, slightly lower than RF and XGBoost but still indicative of its capability to model complex feature interactions. The ANN demonstrated an accuracy of 83.8%, showcasing its ability to capture non-linear relationships despite requiring careful tuning. In contrast, the KNN algorithm recorded the lowest accuracy at 78.9%, reflecting its limitations in high-dimensional datasets and its sensitivity to irrelevant features.

### 3.3. ROC AUC Analysis

The ROC AUC was computed for all models to assess their discriminative power across the five GCS categories. The RF algorithm demonstrated the highest mean ROC AUC at 0.9592, with a low standard deviation of 0.0386, underscoring its reliability and robustness. The ANN followed closely with a mean ROC AUC of 0.9533 (SD: 0.0394), reflecting its capability to learn intricate patterns in the dataset. Similarly, XGBoost achieved a mean ROC AUC of 0.9502 (SD: 0.0405), confirming its strength in clinical predictive tasks.

Figure 1 illustrates the ROC curves for each class predicted by the ANN model, highlighting its high predictive capability with AUC values ranging from 0.89 to 0.99 across the five GCS categories.

Figure 2 illustrates the ROC curves for each class predicted by the ETR model. Notably, the ET model achieved a perfect AUC for class 0 (AUC = 1.00), while performance for class 1 was lower (AUC = 0.74), reflecting variability in its ability to distinguish between certain GCS categories.

Figure 3 illustrates the ROC curves for each class predicted by the KNN model. While the model achieved perfect performance for class 0 (AUC = 1.00), its performance for class 1 (AUC = 0.49) and other classes (e.g., class 3 with AUC = 0.66) was significantly weaker, reflecting its inability to generalize across GCS categories.

Figure 4 depicts the ROC curves for each class predicted by the RF model. The RF model achieved near-perfect performance for all GCS categories, with AUC values ranging from 0.90 to 1.00, further affirming its robustness and suitability for clinical predictive analytics.

Figure 5 displays the ROC curves for each class predicted by the SVM model. Notably, the SVM achieved strong performance for class 0 (AUC = 0.99) and class 2 (AUC = 0.96), while its performance for class 3 (AUC = 0.88) was comparatively lower.

Figure 6 illustrates the ROC curves for each class predicted by the XGBoost model. The model consistently demonstrated excellent performance, with AUC values exceeding 0.90 for most GCS categories, including a perfect AUC of 1.00 for class 0.

The SVM exhibited a mean ROC AUC of 0.9462 (SD: 0.0389), maintaining competitive performance despite its reliance on a linear kernel. The ETR, while achieving a reasonable ROC AUC of 0.9051, exhibited higher variability (SD: 0.1026), which may be attributed to its sensitivity to dataset partitioning. The KNN algorithm displayed the lowest ROC AUC at 0.7472, with a high standard deviation of 0.1913, reflecting its instability and inability to effectively differentiate between outcome categories.

While the overall ROC AUC scores indicate strong model performance in predicting GCS outcomes for AComA aneurysm patients, variability across GCS categories must be considered. The models performed best in extreme cases (GCS <5 and 15), where distinct neurological patterns facilitate classification. In contrast, mid-range GCS scores (9–12) showed lower AUC values, reflecting the clinical heterogeneity of AComA aneurysm patients. Post-operative factors such as delayed cerebral ischemia (DCI), vasospasm, hydrocephalus, and cerebral edema can lead to fluctuating neurological states and overlapping symptoms, making prediction more challenging. These findings align with real-world observations, where moderate GCS scores often correspond to varied recovery trajectories. While ML models effectively distinguish severe and mild cases, their classification accuracy in intermediate scores may be limited by post-operative complications and the subjective nature of neurological assessment. Integrating imaging biomarkers or physiological monitoring could improve predictive performance in these cases.

Furthermore, the DeLong test yielded a *p*-value of 1.00, suggesting that the differences in AUC values between the models are not statistically significant. This outcome aligns with the consistently high and comparable ROC performance observed across all models.

To further illustrate these findings, Figure 1, Figure 2, Figure 3, Figure 4, Figure 5 and Figure 6 provide ROC curves for each model, demonstrating how classification performance varies across different GCS categories. These findings underscore the potential of ML models to revolutionize GCS assessment, particularly in distinguishing clear-cut cases of severe impairment or full recovery. However, the nuanced variability observed in mid-range GCS predictions highlights an opportunity for further refinement. Future advancements integrating multi-modal data sources—such as neuroimaging biomarkers, continuous physiological monitoring, and longitudinal patient trajectories—may enhance model precision, enabling more accurate classification of patients in clinically ambiguous states. These refinements could enhance the reliability of ML models, improving their usefulness in neurological assessment and critical care decision making.

### 3.4. Feature Contributions to Model Performance

The dataset’s comprehensive structure, with its diverse variables, played a critical role in the performance of the ML models. Key pre-surgical features, such as cognitive impairment, memory impairment, and visual disturbances before surgery, were particularly influential, providing early indicators of patient outcomes. Surgical parameters, including aneurysm diameter and Hunt and Hess scores, were also significant, as they directly correlated with the severity of the condition and the complexity of treatment.

Post-surgical outcomes, such as memory impairment after surgery and prolonged intubation, contributed valuable insights into recovery trajectories, allowing the models to identify patterns associated with poor or favorable outcomes. The combination of categorical, continuous, and integer-based features enhanced the models’ ability to capture nuanced interactions, especially for methods like RF and XGBoost.

The superior performance of RF and XGBoost can be attributed to their ability to handle heterogeneous data types, while the slightly lower performance of ANN highlights the challenges of optimizing deep learning architectures for smaller clinical datasets. The relatively poor performance of KNN reflects its sensitivity to the dataset’s high dimensionality and the diverse range of feature types.

### 3.5. Detailed Summary of Results

To evaluate the performance of each machine learning model, we first assessed classification accuracy on the test dataset. Table 2 summarizes these results, highlighting the relative effectiveness of each algorithm in correctly predicting patient outcomes.

Figure 7 provides a comparative visualization of ROC AUC scores, complementing the accuracy metrics detailed in Table 3.

The KNN algorithm exhibited the lowest performance among all models, with an accuracy of 78.9% and a mean ROC AUC of 0.7472, accompanied by the highest standard deviation (0.1913). This suggests that KNN struggled to generalize within a dataset characterized by high dimensionality, heterogeneous feature distributions, and complex interdependencies between variables. Unlike tree-based and boosting models, which inherently capture nonlinear relationships and feature importance, KNN relies solely on distance-based similarity measures, making it vulnerable to noise, irrelevant features, and class imbalances. A key limitation of KNN in this study is its inability to effectively separate mid-range GCS categories (e.g., 9–12), which represent patients with more nuanced and overlapping clinical profiles. Since KNN lacks an internal feature weighting mechanism, it treats all variables equally, disregarding the fact that certain predictors—such as Hunt and Hess scores or post-operative intubation duration—carry significantly more prognostic value than others. Consequently, KNN’s decision boundaries become blurred in cases where neurological deterioration follows complex, multi-dimensional patterns that cannot be easily captured through proximity-based classification.

Additionally, the presence of mixed data types (categorical, ordinal, and continuous variables) further exacerbates KNN’s weaknesses. Distance-based methods, particularly those using Euclidean distance, perform suboptimally in datasets where categorical variables (e.g., presence of cognitive impairments) interact with continuous metrics (e.g., aneurysm diameter). Without proper handling of feature scales and clinical weightings, KNN is prone to misclassification, particularly in intermediate cases where subtle shifts in a few key variables can significantly influence patient trajectories.

In contrast, RF and XGBoost achieved superior results by leveraging feature-driven modeling, ensemble learning, and automated feature selection. These models effectively navigate class imbalances, capture nonlinear interactions, and prioritize critical variables without requiring manual distance metrics. However, despite their strong overall performance, misclassification patterns reveal that errors were disproportionately concentrated in mid-range GCS categories (9–12), where clinical presentations often overlap. The models struggled with fine-grained distinctions rather than broad classification errors, as misclassified cases within this range were predominantly assigned to adjacent categories (e.g., a GCS 10 classified as 8 or 12), rather than extreme misclassifications. This trend reflects the inherent challenges of predicting intermediate neurological states, where small variations in key clinical features—such as post-operative complications or subtle cognitive deficits—can significantly impact GCS classification. Unlike extreme cases (GCS < 5 or 15), where well-defined clinical markers create clear distinctions, mid-range patients often present with overlapping symptomatology, leading to classification ambiguity. RF and XGBoost exhibited higher confidence in extreme cases but greater variability in mid-range classifications, reinforcing the difficulty of distinguishing between patients with moderate impairment.

These findings highlight the importance of not only optimizing model accuracy but also ensuring stability in predictions across clinical subgroups. Identifying areas where misclassifications occur most frequently provides a foundation for refining ML-driven GCS assessments, potentially through the integration of temporal trends, uncertainty quantification, or expanded feature representations that better capture the subtle distinctions within intermediate GCS categories. Future work may explore hybrid methodologies that incorporate domain-specific feature engineering or adaptive weighting mechanisms to further improve classification performance in high-dimensional neurological datasets.

### 3.6. Implications of Findings

The results underscore the importance of feature diversity in clinical predictive modeling. Methods, particularly RF and XGBoost, excelled due to their ability to integrate categorical, continuous, and integer-based features seamlessly. The robust performance of ANN highlights the potential of deep learning in clinical applications, though its sensitivity to hyperparameter tuning suggests further optimization is necessary. Conversely, the poor performance of KNN emphasizes the limitations of simpler methods when faced with high-dimensional, heterogeneous data.

These findings provide valuable insights for the development of clinical decision support tools, demonstrating the power of ML in leveraging diverse patient data to predict outcomes accurately.

## 4. Discussion

### 4.1. Model Performance and Comparative Analysis

This study demonstrates the potential of ML to transform the prediction of GCS scores, a critical measure in neurological assessment and acute care. Among the six evaluated algorithms, RF delivered the best performance, achieving an accuracy of 86.4% and a mean ROC AUC of 0.9592. These results reaffirm the superiority of ensemble methods in managing heterogeneous clinical datasets with mixed variable types, such as categorical, continuous, and integer-based features. The RF model’s ability to aggregate predictions across multiple decision trees allows it to model complex interactions effectively while minimizing variance and bias, ensuring robustness and consistency.

XGBoost, another ensemble-based algorithm, demonstrated similarly strong results (ROC AUC = 0.9502). Its gradient-boosting framework, coupled with regularization, effectively captures intricate feature dependencies while avoiding overfitting. Support vector machines (SVMs) also performed competitively (ROC AUC = 0.9462), underscoring the effectiveness of kernel-based methods in structured datasets. However, SVM’s linear kernel limits its flexibility compared to ensemble models, which may explain its slightly lower performance in this study.

In contrast, KNN struggled significantly, with a mean ROC AUC of 0.7472 and a high standard deviation of 0.1913. KNN’s reliance on proximity-based classification likely hindered its ability to capture the complex relationships inherent in high-dimensional datasets. These findings underscore the importance of selecting advanced ML algorithms, such as ensemble and boosting methods, for modeling intricate clinical data.

While ensemble models like RF and XGBoost demonstrated excellent predictive performance, we acknowledge that their structural complexity—such as the use of max_depth = 180 in XGBoost and the multi-layer architecture of the ANN—may raise concerns about overfitting, particularly in the context of a moderately sized dataset (~759 patients). However, we would like to clarify that our study followed standard machine learning protocols to mitigate this risk. All models were developed using GridSearchCV for hyperparameter tuning, combined with five-fold stratified cross-validation to ensure that parameter selection was based on generalizable performance patterns rather than overfitting to specific data partitions.

Moreover, the test set was completely held out during both training and tuning phases, providing an independent benchmark for model evaluation. We do not consider overfitting to be a major issue, as all top-performing models—including RF, XGBoost, SVM, and ANN—achieved robust predictive performance, with accuracy scores exceeding 80% and ROC AUC values consistently above 0.94 on the unseen test data. These results suggest that the models generalized well within the studied population.

That said, we acknowledge that the dataset was slightly imbalanced across certain GCS classes, particularly in the mid-range categories (e.g., GCS 9–12). This imbalance may have limited the models’ ability to fully capture distinguishing features for underrepresented groups and likely contributed to the misclassification patterns observed in those categories. Additionally, the dataset was derived from a single institution, which may constrain the generalizability of the models to external populations with different clinical workflows, demographic characteristics, or documentation standards.

Therefore, while our findings support the feasibility and clinical value of ML-based GCS prediction, future work should prioritize external validation using multi-institutional datasets and explore model simplification techniques to ensure that predictive performance remains stable and interpretable across diverse clinical settings.

When compared to existing literature, this study fills a critical gap by focusing specifically on GCS prediction, a relatively underexplored area in ML research. Previous studies predominantly target broader outcomes, such as mortality or long-term functional recovery, leaving the unique challenges of GCS prediction inadequately addressed [14,15]. By tailoring ML models to this metric, our work provides a novel approach that is directly applicable to acute care decision making.

### 4.2. Feature Contributions and Interpretability

A cornerstone of this study is the identification of key clinical features driving GCS predictions. Variables such as the Hunt and Hess score, aneurysm diameter, and pre-operative cognitive impairments emerged as critical predictors, aligning with well-established clinical paradigms. The Hunt and Hess score, a validated measure of aneurysm severity, remains an indispensable tool for stratifying patients based on their risk of complications. Its prominence in this study reinforces its role as a reliable indicator of neurological outcomes.

Post-operative variables, including prolonged intubation and memory impairment, added a dynamic perspective to the predictive models. These features reflect the complexities of recovery trajectories and highlight the importance of incorporating temporal data to account for changes in patient status. The integration of features spanning pre-, intra-, and post-operative phases ensures that the models provide a holistic assessment of GCS outcomes, moving beyond static scoring to dynamic, data-driven predictions.

A significant strength of this study is its emphasis on model interpretability. Ensemble methods like RF and XGBoost not only deliver high accuracy but also offer transparency through feature importance metrics. By elucidating the contribution of individual variables to predictions, these models foster clinician trust and enable actionable insights. For instance, a clinician could use the model to identify high-risk factors, such as a severe Hunt and Hess score or significant post-operative complications, and tailor interventions accordingly. This interpretability bridges the gap between computational advancements and clinical usability, ensuring that predictions are not perceived as opaque or arbitrary.

Beyond risk identification, leveraging ML-derived insights for personalized intervention strategies remains a crucial next step. The ability to quantify how specific clinical variables influence GCS outcomes enables a more proactive approach to patient management, where individualized care pathways can be developed based on data-driven predictions. For example, recognizing subtle pre-operative risk factors could facilitate earlier intervention planning, while post-operative predictors may support real-time adjustments to rehabilitation protocols based on expected recovery trajectories. Dynamic risk stratification models, integrating continuously updated patient data, could further refine these assessments, ensuring that clinical decisions evolve alongside a patient’s condition. To maximize clinical impact, future implementations should prioritize seamless integration into existing workflows, ensuring that ML-generated insights are delivered in a format that is interpretable, actionable, and aligned with current neurological assessment frameworks. Developing interactive dashboards or real-time decision-support tools could enhance usability, allowing clinicians to efficiently incorporate predictive insights into critical care management.

### 4.3. Novelty and Relevance to the Field

This study represents a novel application of ML in neurological care by targeting GCS prediction, a critical yet underexplored area. Unlike previous research, which often focuses on broader or aggregated outcomes, this work delves into the nuanced assessment of consciousness levels. GCS predictions are particularly valuable in acute care settings, where timely and precise assessments guide triage decisions, surgical planning, and resource allocation.

Additionally, the emphasis on interpretability distinguishes this study from prior ML applications in healthcare. By using models like RF and XGBoost, this work ensures that clinicians can understand and trust the factors driving predictions. This transparency is crucial for adoption in real-world settings, where black-box algorithms often face resistance due to their lack of explainability. By prioritizing both accuracy and usability, this study bridges the gap between technological innovation and clinical applicability.

The methodologies employed here are also relevant to the broader field of precision medicine. Ensemble methods have been successfully applied in predicting stroke recovery, post-surgical complications, and critical care outcomes. By extending these techniques to GCS prediction, this study demonstrates their versatility and effectiveness, providing a foundation for further advancements in neurological care.

### 4.4. Broader Applications and Clinical Implications

The findings of this study extend beyond GCS prediction, offering valuable insights into the broader applicability of ML in neurological and critical care. The methodologies demonstrated here can be adapted to a wide range of clinical scenarios, including:Predicting functional recovery in stroke patients.Assessing severity and treatment needs in traumatic brain injuries.Forecasting complications in post-surgical care.

Integrating these models into triage workflows in emergency departments could significantly enhance the prioritization of high-risk patients, ensuring timely interventions and optimal resource allocation.

In clinical practice, the ability to provide accurate and interpretable GCS predictions has profound implications for patient care. For example, patients predicted to have low GCS scores can be prioritized for aggressive interventions, such as neurosurgical procedures or intensive monitoring, while those with higher predicted scores may benefit from conservative management strategies. This dynamic, data-driven approach improves both short-term outcomes, such as survival rates and neurological recovery, and long-term care strategies, including rehabilitation planning and quality-of-life considerations.

### 4.5. Ethical Considerations

The integration of ML into healthcare, particularly in high-stakes environments like critical care, raises several ethical considerations that must be addressed to ensure equitable and responsible implementation.

#### 4.5.1. Bias and Fairness in Predictions

The risk of algorithmic bias is a key concern, particularly when demographic variables such as age, sex, or ethnicity are included in the dataset. While these features may enhance predictive accuracy, they can also inadvertently propagate existing healthcare disparities. For instance, differences in healthcare access or comorbidity prevalence across populations could skew the model’s outputs, disadvantaging certain groups. Ensuring fairness requires rigorous evaluation of model performance across diverse subpopulations and the implementation of bias mitigation strategies [16].

#### 4.5.2. Transparency and Accountability

Transparency in model predictions is critical for fostering trust among clinicians and patients. Some algorithms inherently offer interpretability through feature importance metrics, allowing clinicians to understand the factors driving predictions. However, continuous efforts to improve explainability, such as integrating SHAP values or decision path visualizations, are essential to maintain accountability in clinical decisions [17,18].

#### 4.5.3. Augmenting, Not Replacing, Clinical Expertise

ML models should complement, not replace, clinical judgment. While these tools provide valuable insights, final decisions must remain in the hands of healthcare professionals who can contextualize predictions within the broader clinical picture. Ensuring that models are used as augmentative tools rather than standalone decision-makers is critical to maintaining ethical standards in patient care [19].

#### 4.5.4. Data Privacy and Security

Protecting patient data is paramount as ML models are integrated into clinical workflows. The use of federated learning frameworks, which allow models to be trained across multiple institutions without sharing raw data, offers a promising solution to enhance privacy while improving model generalizability. Compliance with data protection regulations, such as GDPR and HIPAA, must be rigorously enforced to maintain public trust in these technologies [20].

Beyond these considerations, another challenge lies in the transition from model development to real-time deployment in clinical environments. While the models performed well on retrospective data, adapting them for prospective, real-time use requires addressing issues such as computational efficiency, latency, and seamless integration with existing hospital infrastructures. Future work should focus on optimizing inference speed to ensure that predictions are generated in clinically relevant time frames without disrupting workflow efficiency. Additionally, bridging the gap between predictive outputs and actionable clinical decisions remains an ongoing challenge. Even with strong model performance, the ability to provide clear, interpretable recommendations tailored to individual patients is crucial for clinician trust and adoption. Developing intuitive decision-support interfaces that present ML-driven insights in a way that aligns with existing neurological scoring frameworks could improve usability and acceptance [21].

Lastly, continuous model updating and adaptation to evolving clinical practices must be considered. As medical guidelines change and new data sources become available, ensuring that ML models remain relevant will require ongoing validation, recalibration, and feedback loops that incorporate real-world outcomes. Establishing mechanisms for periodic retraining and performance monitoring will be essential to maintain reliability over time.

### 4.6. Future Technological Directions

Advancing GCS prediction further requires incorporating additional data modalities, such as imaging (CT/MRI), genomic information, and real-time monitoring systems [22]. These multi-modal datasets could significantly enhance the precision of predictions, providing a comprehensive view of patient status. Emerging technologies like explainable AI (XAI) could further improve interpretability, enabling clinicians to validate predictions with greater confidence [23].

Federated learning, which facilitates multi-institutional collaborations while preserving patient privacy, holds promise for addressing the limitations of single-center datasets. Additionally, incorporating uncertainty quantification, such as confidence intervals or probabilistic outputs, could improve the reliability of ML tools in high-stakes environments [24]. Beyond expanding data modalities, the next frontier in GCS prediction lies in the fusion of real-time adaptive learning with patient-specific trajectory modeling. Current ML models, while effective, remain limited by static training paradigms that do not evolve with incoming patient data. The integration of self-learning systems that continuously refine their predictions based on new clinical inputs could revolutionize neurological assessment, enabling real-time model recalibration without requiring manual retraining.

Additionally, the incorporation of digital twin technology could further refine predictive accuracy by creating individualized, AI-driven patient models that dynamically adjust based on physiological and clinical changes. These virtual representations could simulate various therapeutic scenarios, allowing clinicians to explore potential intervention outcomes before implementing them in real time. Combining digital twin simulations with ML-powered risk stratification could pave the way for precision medicine in neurocritical care [25].

To support effective deployment, future research should explore ways to improve model adaptability through continual learning strategies, allowing predictive performance to be monitored and updated based on longitudinal patient outcomes. Developing cloud-based, decentralized ML infrastructures may also help facilitate real-time computation and integration into intensive care settings. As these technologies advance, improving the connection between predictive analytics and clinical decision support could enhance the role of ML in GCS assessment and neurological care [26].

Beyond structural imaging and adaptive learning, the integration of electroencephalography (EEG) represents a pivotal advancement in GCS prediction models, offering real-time insights into cortical function. Unlike static imaging modalities, which capture a snapshot of brain pathology, EEG provides continuous, high-resolution data on neuronal activity, enabling early detection of subclinical seizures, non-convulsive status epilepticus, and global cortical dysfunction—factors that significantly influence consciousness levels but may be overlooked in conventional assessments [27].

ML-driven EEG analysis could refine GCS prediction by distinguishing between primary neurological impairment and reversible metabolic or pharmacologic influences on consciousness. For example, machine learning models trained on EEG-derived features—such as spectral power distribution, burst suppression patterns, and phase-amplitude coupling metrics—could enhance diagnostic precision in patients where GCS alone provides limited prognostic value [28]. Automated feature extraction through convolutional neural networks (CNNs) and temporal sequence modeling via long short-term memory (LSTM) networks or transformers could enable predictive systems to detect subtle electrophysiological signatures of impending deterioration far earlier than clinical assessment alone [29].

Moreover, EEG’s bedside accessibility makes it an ideal candidate for integration into real-time adaptive ML frameworks. Continuous monitoring allows models to recalibrate dynamically as a patient’s neurological state evolves, a crucial advantage in conditions where GCS fluctuations occur due to transient metabolic derangements or sedation effects. Future research should explore fusion models that integrate EEG time-series data with neuroimaging biomarkers and clinical parameters, creating a multi-modal system capable of contextualizing consciousness impairment within a broader neurophysiological landscape [30].

To ensure the clinical viability of EEG-enhanced ML models, standardization of acquisition protocols, development of artifact-resistant algorithms, and validation across diverse patient cohorts will be essential. Prospective studies leveraging federated learning approaches could facilitate large-scale, privacy-preserving model training across multiple institutions, paving the way for AI-driven, precision neuromonitoring in critical care settings [31].

Beyond traumatic brain injury, ML-driven neurological assessment could be extended to ischemic and hemorrhagic stroke, integrating multiple clinical scales for a more comprehensive prognosis. The GCS, while effective in assessing global consciousness levels, does not capture focal neurological deficits, which are central to stroke evaluation. The National Institutes of Health Stroke Scale (NIHSS) provides a more granular assessment of motor, sensory, and cognitive impairments in stroke patients, making it a complementary tool for ML-based predictive models [32].

Integrating GCS and NIHSS within an ML framework could improve stratification of stroke severity and functional outcomes, especially when combined with neuroimaging and biochemical biomarkers. Advanced models incorporating C-reactive protein, D-dimer levels, and hypoxia markers could enhance the prediction of secondary complications such as post-stroke edema, hemorrhagic transformation, or delayed neurological deterioration. Deep learning architectures, particularly fusion models that combine structured clinical data, imaging-derived features, and biochemical trends, could provide a more precise risk assessment in acute stroke care [33].

## 5. Limitations

While this study represents a significant advancement in the application of ML for predicting GCS scores, several limitations must be acknowledged to provide context and guide future research.

The dataset used, though comprehensive and rich in clinical features, was derived from a single center and consists predominantly of ruptured AComA aneurysms. This specific patient population may limit the generalizability of the findings to other aneurysm types or unruptured cases. External validation with multicenter datasets, including a broader range of aneurysm presentations, is essential to confirm the robustness and applicability of these models across diverse healthcare settings.

Additionally, while the dataset included a wide range of clinical variables, certain valuable data types, such as imaging (e.g., CT/MRI) and biochemical markers, were not available. Incorporating these modalities in future work could further enhance predictive accuracy and provide deeper insights into neurological outcomes.

A challenge observed was the higher misclassification rates in intermediate GCS categories (e.g., scores between 9 and 12). This reflects the overlapping clinical features within these groups, underscoring the need for additional granular data or temporal information to better distinguish such cases.

While RF and XGBoost provided interpretable predictions through feature importance metrics, the interpretability could be further refined using advanced tools like SHAP to make the insights more accessible and actionable for clinicians.

Finally, implementing these models in real-world clinical workflows will require thoughtful integration into electronic health records (EHRs), robust validation across populations, and clinician training to ensure seamless adoption.

Despite these limitations, this study establishes a strong foundation for ML-driven GCS prediction, combining high accuracy with clinical relevance. It exemplifies the potential of advanced algorithms to enhance critical care decision making, paving the way for future innovations in neurological assessment and personalized medicine.

Although GCS is widely used in neurological assessment, its predictive accuracy may vary depending on the underlying pathology. Conditions such as stroke, metabolic encephalopathy, and intracranial infections differ from traumatic brain injury (TBI) in their mechanisms of consciousness impairment, which could introduce biases in ML-based predictions. In metabolic encephalopathy, fluctuations in mental status may result from biochemical imbalances rather than structural injury, making GCS scores less stable indicators of neurological function.

Similarly, in brainstem strokes, impaired motor and verbal responses may not accurately reflect overall consciousness, potentially affecting model performance. These variations highlight the need for careful consideration when applying GCS-driven models across diverse patient groups. Future advancements should focus on integrating complementary assessment tools such as the NIH Stroke Scale for cerebrovascular events or biochemical markers for metabolic disturbances. Additionally, prospective validation in heterogeneous patient populations will be essential to refine model generalizability and ensure clinical reliability across different neurological conditions.

## 6. Conclusions

This study represents a significant step forward in the application of ML for GCS prediction, offering a novel approach to tackling one of neurology’s most critical challenges. By employing advanced ML algorithms and integrating a diverse set of clinical features, this work provides a new paradigm for improving outcome prediction in critical care settings.

Our models demonstrated exceptional predictive performance while maintaining interpretability, a key requirement for clinical adoption. These tools go beyond static scoring systems, offering a dynamic, data-driven framework that aligns with the evolving needs of modern critical care.

Crucially, this study underscores the importance of clinically relevant features in driving accurate predictions. By incorporating a comprehensive dataset and leveraging pre-, intra-, and post-operative variables, we have paved the way for predictive tools that are both precise and contextually meaningful. This focus on interpretability ensures that clinicians can trust and rely on these models, fostering a bridge between computational innovation and practical implementation.

The implications of this work extend beyond GCS prediction. The methodologies and insights developed here are adaptable to other areas of neurological care, including the assessment of stroke outcomes, traumatic brain injuries, and post-surgical recovery. These findings highlight the transformative potential of ML in advancing personalized medicine and improving decision making across the continuum of care.

Future efforts should build on this foundation by incorporating additional data modalities, such as imaging and temporal trends, and validating these models across diverse healthcare systems. By addressing these opportunities, we can further enhance the precision, generalizability, and clinical impact of ML-driven tools.

In conclusion, this study exemplifies the potential of ML to redefine how we approach critical neurological assessments. It not only provides a framework for improving patient outcomes but also establishes a pathway for integrating intelligent, interpretable models into the heart of clinical practice. As we move toward a more data-driven era of medicine, this work serves as a testament to the power of combining scientific innovation with clinical relevance to create meaningful change in healthcare.

## Figures and Tables

**Figure 1 jcm-14-02672-f001:**
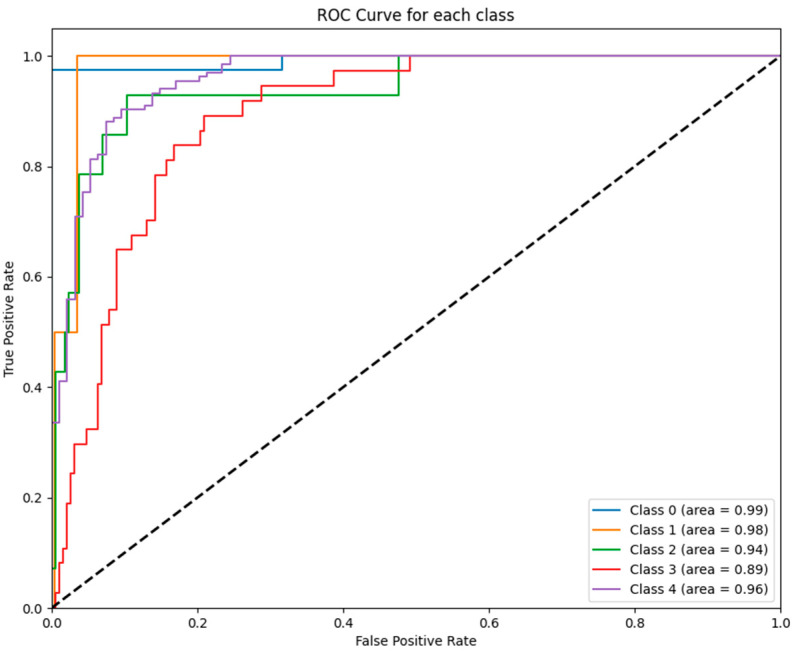
ROC curves for ANN model: The figure displays the ROC curves for each GCS category predicted by the ANN model. The AUC values demonstrate excellent performance across all classes, ranging from 0.89 (class 3) to 0.99 (class 0).

**Figure 2 jcm-14-02672-f002:**
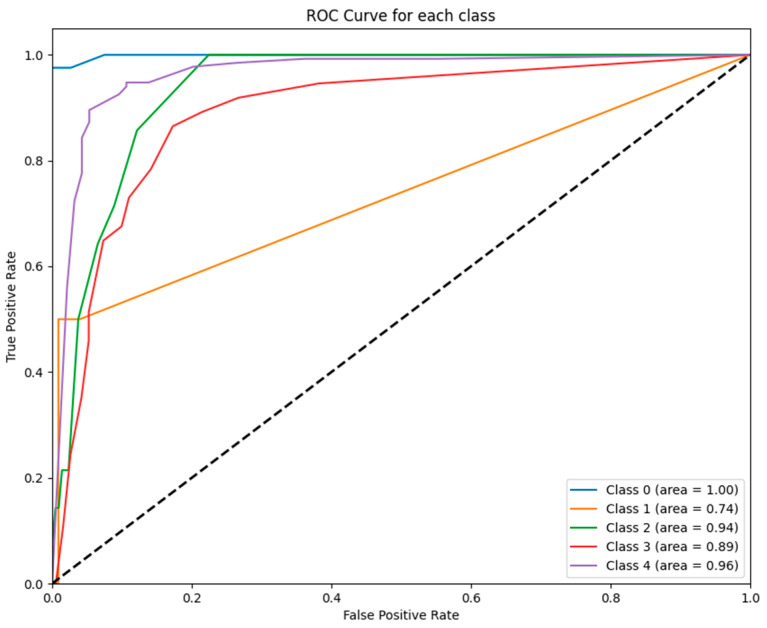
ROC curves for ETR model: This figure displays the ROC curves for each GCS category as predicted by the ET model. The model performed exceptionally well for class 0 (AUC = 1.00) and class 2 (AUC = 0.94) but exhibited weaker differentiation for class 1 (AUC = 0.74).

**Figure 3 jcm-14-02672-f003:**
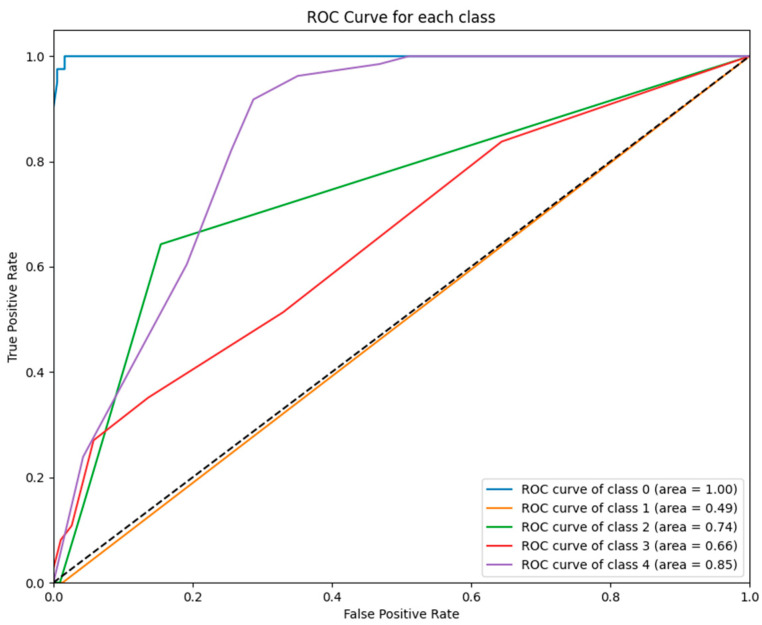
ROC curves for KNN model: The figure shows the ROC curves for each GCS category predicted by the KNN model. While class 0 achieved a perfect AUC (1.00), performance was significantly weaker for class 1 (AUC = 0.49) and class 3 (AUC = 0.66), highlighting the model’s variability in classifying GCS outcomes.

**Figure 4 jcm-14-02672-f004:**
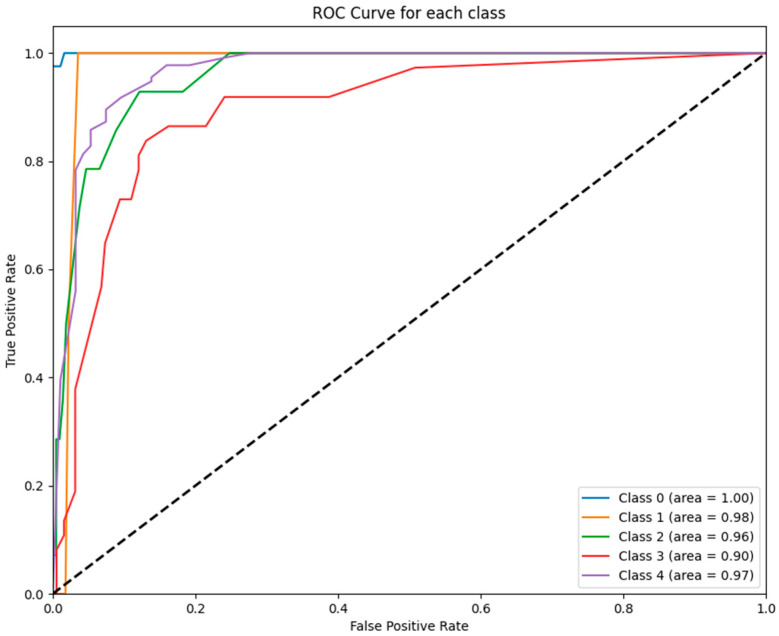
ROC curves for RF model: The figure displays the ROC curves for each GCS category as predicted by the RF model. The AUC values reflect exceptional performance across all classes, ranging from 0.90 (class 3) to 1.00 (class 0).

**Figure 5 jcm-14-02672-f005:**
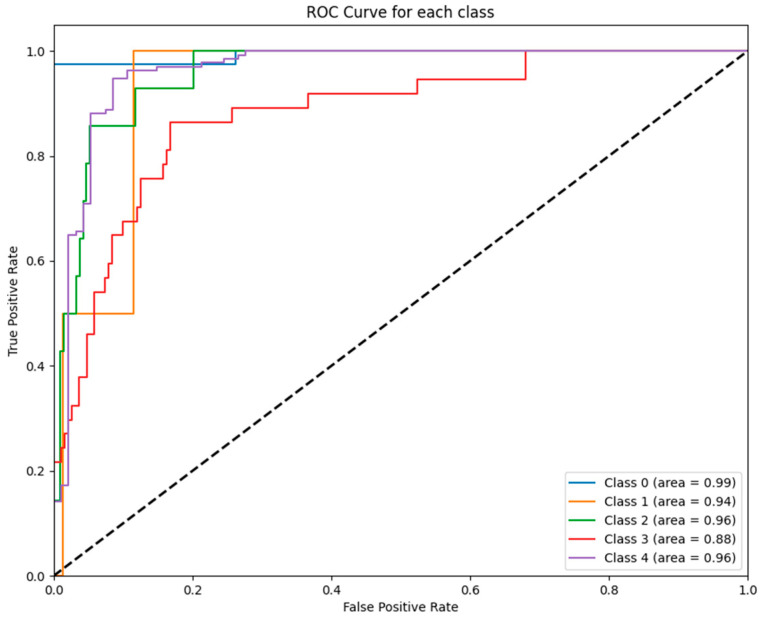
ROC curves for SVM Model: The figure shows the ROC curves for each GCS category predicted by the SVM model. High AUC values were observed for class 0 (0.99), class 1 (0.94), and class 2 (0.96), with slightly lower performance for class 3 (0.88).

**Figure 6 jcm-14-02672-f006:**
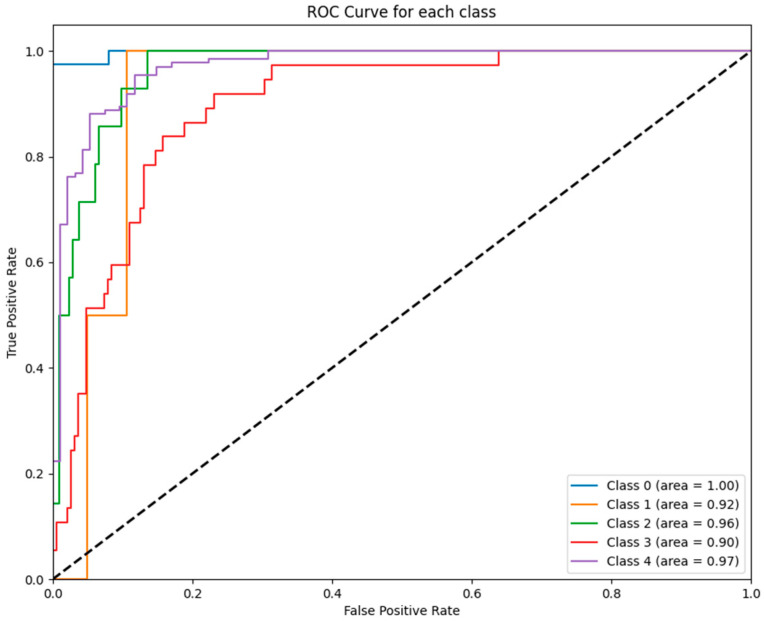
ROC curves for XGBoost model: The figure shows the ROC curves for each GCS category as predicted by the XGBoost model. The AUC values highlight exceptional performance, ranging from 0.90 (class 3) to 1.00 (class 0), demonstrating the model’s robustness in multi-class classification.

**Figure 7 jcm-14-02672-f007:**
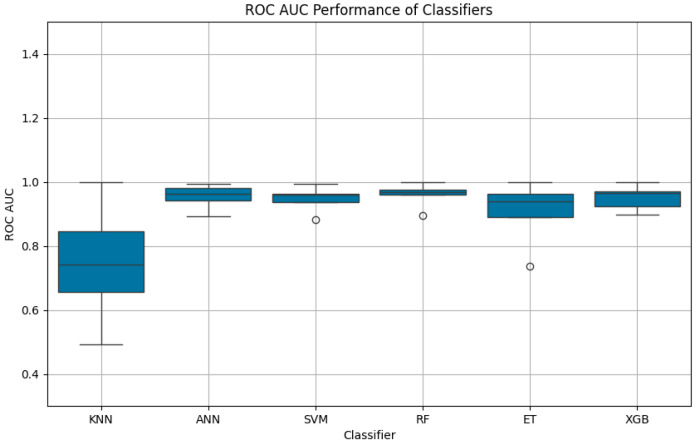
Boxplot of ROC AUC performance for classifiers: The figure illustrates the mean ROC AUC scores and their variability for six machine learning algorithms. RF, ANN, and XGBoost (XGB) exhibited high consistency and superior performance, while KNN displayed the lowest mean ROC AUC and the highest variability.

**Table 1 jcm-14-02672-t001:** Demographic characteristics of the 759-patient study cohort. The table includes gender and residence type distributions, as well as estimated age group counts. Patients are evenly split by gender and urban/rural background, and the most represented age group is 51–60 years. Age distribution by gender shows a slight female predominance in older age groups, while urban and rural patterns appear similar, supporting the dataset’s balance across key demographic factors.

Variable	Category	Count	Percentage (%)
Gender	Female	400	52.7
	Male	359	47.3
Residence type	Rural	396	52.2
	Urban	363	47.8
Age group	20–30 years	~30	~4.0
	31–40 years	~70	~9.2
	41–50 years	~120	~15.8
	51–60 years (peak)	~160	~21.1
	61–70 years	~140	~18.4
	71–80+ years	~100	~13.2
Age by gender	Older females	-	Slight predominance in 60–80+ age groups
Age by residence	Urban vs. rural	-	Similar distribution across all age groups

**Table 2 jcm-14-02672-t002:** Accuracy on the Test Dataset. This table summarizes the accuracy performance of six machine learning algorithms on the test dataset. The accuracy metric reflects the percentage of correct predictions made by each model. The RF algorithm achieved the highest accuracy (86.4%), outperforming all other models.

Algorithm	Accuracy (%)
Random forest (RF)	86.4
Support vector machine	85.9
XGBoost	85.9
Extra trees regressor	85.1
Artificial neural network	83.8
K-nearest neighbors	78.9

**Table 3 jcm-14-02672-t003:** Mean ROC AUC and Standard Deviation. This table provides the mean ROC AUC (receiver operating characteristic area under the curve) scores and their corresponding standard deviations for the six machine learning models. The ROC AUC metric evaluates the models’ ability to distinguish between the five GCS categories. The RF model achieved the highest mean ROC AUC (0.9592) with a low standard deviation (0.0386), indicating both superior predictive power and consistency.

Algorithm	Mean ROC AUC	Standard Deviation
Random forest (RF)	0.9592	0.0386
Artificial neural network	0.9533	0.0394
XGBoost	0.9502	0.0405
Support vector machine	0.9462	0.0389
Extra trees regressor	0.9051	0.1026
K-nearest neighbors	0.7472	0.1913

## Data Availability

Available upon reasonable request from the corresponding author.

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
