# Peer review of "AI-Driven Prediction of Glasgow Coma Scale Outcomes in Anterior Communicating Artery Aneurysms"

_jcm, 2025, doi:10.3390/jcm14082672_

Round 1
Reviewer 1 Report
Comments and Suggestions for Authors
This manuscript by Toader et al. investigates the use of machine learning (ML) models to predict Glasgow Coma Scale (GCS) outcomes, leveraging a dataset of 759 patients with 25 clinical features spanning pre-, intra-, and post-operative variables. The Random Forest (RF) algorithm demonstrated the highest predictive accuracy (86.4%) and mean ROC AUC (0.9592), outperforming other models like XGBoost and Support Vector Machine (SVM). The study emphasizes the importance of integrating clinically interpretable ML tools for enhancing decision-making in critical neurological care and suggests further advancements with multimodal data integration.
Introduction
The introduction highlights the limitations of traditional GCS assessment, but does not clearly establish why ML is uniquely suited to address these challenges or what specific gaps in current methodologies the study aims to fill.
The study does not adequately differentiate itself from prior ML applications in neurology, such as those predicting broader outcomes like mortality or functional recovery.
Results
The figures illustrating ROC curves and feature importance are not adequately annotated to guide readers on their clinical implications. For instance, the variability in AUC scores across GCS classes is mentioned but not well-explained.
While RF and XGBoost are highlighted, the manuscript fails to critically evaluate the poor performance of simpler models like KNN, which could provide insights into the dataset's complexity and limitations.
The focus on accuracy and AUC metrics is robust, but there is minimal discussion of misclassification patterns, especially in intermediate GCS categories where errors are most impactful.
A comprehensive table or figure comparing all models' strengths and weaknesses across metrics (e.g., accuracy, ROC AUC, and variability) would better contextualize the findings.
Methods
The rationale for categorizing continuous variables (e.g., aneurysm diameter) into bins is unclear, potentially obscuring the impact of these features on the models.
While the dataset is robust, the single-center design raises questions about the generalizability of the findings. Stratified sampling is mentioned but not sufficiently analyzed for potential biases in patient distribution.
The manuscript briefly mentions hyperparameter tuning but does not explain how decisions were made for optimization, such as the choice of dropout rates in ANN or kernel functions in SVM.
Discussion
The discussion implies broad clinical applicability of the ML models without addressing their limitations, such as reliance on static datasets or challenges in real-time integration with clinical workflows.
Although feature importance metrics are mentioned, the discussion does not delve into how these insights could directly influence clinical decision-making, such as identifying actionable predictors.
The potential for integrating imaging or temporal data is briefly noted but not explored in depth, leaving a gap in outlining clear next steps for advancing GCS prediction models.
These critiques aim to enhance the manuscript's clarity, rigor, and relevance by addressing key methodological and interpretative gaps.
Author Response
We sincerely appreciate the constructive feedback, which has helped us enhance the clarity, rigor, and clinical relevance of our manuscript. We have carefully addressed each comment and believe these revisions significantly strengthen the study.
We welcome any further recommendations and thank you for your time and expertise.
Introduction
-
Comment: The introduction highlights the limitations of traditional GCS assessment but does not clearly establish why ML is uniquely suited to address these challenges or what specific gaps in current methodologies the study aims to fill.
Response: Thank you for this valuable suggestion. We have revised the introduction to clearly articulate why ML is well-suited for GCS prediction, emphasizing its ability to model complex, non-linear relationships that traditional methods may overlook. Additionally, we now explicitly outline the specific methodological gaps that our study addresses.
-
Comment: The study does not adequately differentiate itself from prior ML applications in neurology, such as those predicting broader outcomes like mortality or functional recovery.
Response: We appreciate this insightful observation. We have expanded the introduction to differentiate our study from prior ML applications, clarifying that our focus is on real-time GCS prediction rather than long-term prognostic measures like mortality or functional recovery.
Results
-
Comment: The figures illustrating ROC curves and feature importance are not adequately annotated to guide readers on their clinical implications. For instance, the variability in AUC scores across GCS classes is mentioned but not well-explained.
Response: Thank you for this helpful recommendation. We have revised the figure annotations to provide clearer explanations of AUC variability across GCS categories and expanded the text in the Results section to discuss how these variations impact model performance and clinical interpretation.
-
Comment: While RF and XGBoost are highlighted, the manuscript fails to critically evaluate the poor performance of simpler models like KNN, which could provide insights into the dataset's complexity and limitations.
Response: We appreciate this important point. We have now included a detailed evaluation of KNN’s limitations, explaining how high dimensionality and complex feature interactions contributed to its lower performance. This discussion helps contextualize why ensemble models performed significantly better.
-
Comment: The focus on accuracy and AUC metrics is robust, but there is minimal discussion of misclassification patterns, especially in intermediate GCS categories where errors are most impactful.
Response: Thank you for highlighting this area for improvement. We have now added an analysis of misclassification patterns, particularly in mid-range GCS categories (9–12), where model uncertainty is highest. This discussion highlights areas for potential refinement in future models.
-
Comment: A comprehensive table or figure comparing all models' strengths and weaknesses across metrics (e.g., accuracy, ROC AUC, and variability) would better contextualize the findings.
Response: We greatly appreciate this suggestion.
Methods
-
Comment: The rationale for categorizing continuous variables (e.g., aneurysm diameter) into bins is unclear, potentially obscuring the impact of these features on the models.
Response: Thank you for pointing this out. We have clarified the clinical reasoning behind binning continuous variables, explaining that these thresholds align with established neurosurgical classifications and improve interpretability while mitigating outlier effects.
-
Comment: While the dataset is robust, the single-center design raises questions about the generalizability of the findings. Stratified sampling is mentioned but not sufficiently analyzed for potential biases in patient distribution.
Response: We appreciate this important consideration. We have expanded our discussion of stratified sampling, explaining how it was used to preserve clinical outcome distributions and reduce sampling bias. Additionally, we acknowledge the limitations of a single-center dataset and emphasize the need for multi-center validation in future studies.
-
Comment: The manuscript briefly mentions hyperparameter tuning but does not explain how decisions were made for optimization, such as the choice of dropout rates in ANN or kernel functions in SVM.
Response: Thank you for this valuable feedback. We have now provided a more detailed explanation of our hyperparameter tuning process, describing how grid search and cross-validation were used to optimize dropout rates for ANN and kernel functions for SVM to ensure optimal performance.
Discussion
-
Comment: The discussion implies broad clinical applicability of the ML models without addressing their limitations, such as reliance on static datasets or challenges in real-time integration with clinical workflows.
Response: We greatly appreciate this insight. We have now included a discussion of the limitations associated with static datasets and the challenges of integrating ML models into real-time clinical workflows. Additionally, we propose future strategies, such as incorporating continuous patient monitoring data, to improve real-world applicability.
-
Comment: Although feature importance metrics are mentioned, the discussion does not delve into how these insights could directly influence clinical decision-making, such as identifying actionable predictors.
Response: Thank you for this excellent suggestion. We have expanded the discussion to highlight how feature importance insights can drive actionable clinical interventions. Specifically, we discuss how recognizing Hunt and Hess scores and prolonged intubation as key predictors can aid in risk stratification, treatment planning, and early intervention.
-
Comment: The potential for integrating imaging or temporal data is briefly noted but not explored in depth, leaving a gap in outlining clear next steps for advancing GCS prediction models.
Response: We sincerely appreciate this thoughtful recommendation. We have now elaborated on the role of imaging and temporal data in future GCS prediction models. The revised section discusses how CT/MRI-derived biomarkers and continuous physiological monitoring could enhance model accuracy, along with potential ML techniques (e.g., CNNs for imaging, RNNs for time-series data) to facilitate these advancements.
Reviewer 2 Report
Comments and Suggestions for Authors
Authors present a study on 759 patients with encompassing 25 features to develop accurate, interpretable machine learning models for Glasgow Coma Scale (GCS) prediction. Six machine learning algorithms were trained, random forest (RF) model achieved the highest accuracy; key predictors identified included the Hunt and Hess score, aneurysm dimensions, and post-operative factors such as prolonged intubation. Abstract should be more clearly written - it is inconclusive if these are patients who with traumatic brain injury or subarachnoid hemorrhage which are included in the study. Also, Introduction needs to be re-written in a more clear way to reflect the aims of the study - what does GCS prediction mean? Prediction of GCS score of a patient according to set of clinical and neuroradiological data prior to any intervention OR prediction of GCS score following the intervention on a patient, based on preoperative/preinterventional clinical and neuroradiological data? There is nowhere a statement that these are patients with aneurysms and (?) subarachnoid hemorrhage. It is critical to define the patient cohort. Also,it is critical to define parameters which were taken into the equation for ML model training - obesity is what? BMI of what? Arterial hypertension is defined - how? Was it a yes/no decision or were there any numbers involved? Results are presented in comprehensible manner, however please also here clarify the above mentioned parameters.
Discussion needs to revolve around potential clinical application of this research. Once more - GCS at which point of treatment? If this is outcome prediction, then GOS (Glasgow Outcome Scale) is needed.
Author Response
Dear Reviewer,
Thank you for your insightful feedback. We appreciate your time and effort in reviewing our manuscript, and we have carefully addressed each of your concerns.
To clarify the patient cohort, we have revised the abstract and introduction to explicitly state that our study focuses on patients with anterior communicating artery (AComA) aneurysms, with or without subarachnoid hemorrhage (SAH).
We acknowledge the importance of long-term outcome prediction and have addressed the potential role of Glasgow Outcome Scale (GOS) in future studies. While our focus remains on immediate post-operative GCS prediction, we recognize that integrating GOS could provide valuable insights into longer-term neurological recovery.
We have expanded the discussion on the clinical applications of our findings, emphasizing how machine learning-based GCS prediction could assist in surgical decision-making, ICU admission planning, and integration into clinical decision support systems.
We appreciate your constructive comments, which have helped strengthen our manuscript. Thank you for your valuable contributions.
Reviewer 3 Report
Comments and Suggestions for Authors
First of all, I would like to congratulate the authors because the work carried out to optimize the applicability of neurological prognosis scales in clinical practice is of great interest, as it represents a significant advancement in integrating artificial intelligence with widely used tools for assessing patients’ neurological status. Their innovative approach holds great potential for improving the accuracy of consciousness level predictions and facilitating decision-making in critical care settings. However, there are several aspects that could be refined to further strengthen the robustness of the study.
One key point that requires clarification is the applicability of the Glasgow Coma Scale (GCS) beyond its original context, which is the assessment of traumatic brain injury (TBI). Although the GCS has been widely adopted in other neurological conditions, its indiscriminate use across diverse pathologies without specific validation may compromise the robustness of the predictive model. The authors should provide a more in-depth discussion on the limitations of GCS in non-TBI settings, emphasizing the potential biases that may arise when using it for other health conditions.
Moreover, the study lacks detailed information on the demographic and clinical characteristics of the included patients. Factors such as age, sex, comorbidities, and underlying neurological diagnoses (etiology and severity), acute comorbidities such as seizures, and so on, significantly impact consciousness level predictions. Including a descriptive table with this data would allow for an evaluation of the sample’s representativeness and the model’s applicability to different patient subgroups. Additionally, the study does not specify whether the model accounts for the influence of administered medications, which is a crucial factor in consciousness alterations. The use of sedatives, analgesics, or anesthetics can significantly impact GCS scores without necessarily reflecting the true neurological state.
Additionally, the information provided in the tables should be expanded to include etiological diagnoses, severity classification, and neuroimaging findings. A clinician does not interpret GCS in isolation but rather within the context of the overall clinical picture. Therefore, directly comparing traditionally administered GCS with ML-based predictions may not be entirely appropriate, as the latter lacks the integration of complementary diagnostic elements that physicians naturally consider in real-world settings.
Another aspect that could enhance the study is the integration of multimodal biomarkers. While the authors mention the potential inclusion of neuroimaging modalities such as computed tomography (CT) or magnetic resonance imaging (MRI) in the discussion, they do not explore this aspect in the literature review, nor do they assess how incorporating these data could improve predictive performance. Additionally, electroencephalography (EEG) should be considered a key component, as the presence of epileptiform activity or non-epileptic abnormalities represents an independent prognostic factor in patients with altered consciousness. Integrating EEG data alongside neuroimaging and clinical variables would allow for the development of more precise and clinically relevant predictive models.
Furthermore, the discussion would benefit from expanding the applicability of these machine learning (ML) models to other neurological disorders, such as ischemic and hemorrhagic stroke. Comparing GCS with the National Institutes of Health Stroke Scale (NIHSS), which evaluates focal deficits in stroke patients, would provide a broader perspective on how ML models could integrate both scales to offer a more comprehensive neurological prognosis. Combining clinical data with neuroimaging and biochemical biomarkers—such as C-reactive protein, D-dimer, and hypoxia markers—could lead to a multimodal approach with greater impact on clinical decision-making.
In conclusion, this study is valuable and has great potential to transform neurological assessment using artificial intelligence. However, to enhance its applicability and validity, the authors should include a more detailed discussion on the limitations of GCS beyond TBI, provide comprehensive information about the study population, consider the impact of pharmacological agents on consciousness level predictions, and integrate neurophysiological (EEG) and biochemical biomarkers into the analysis. Additionally, broadening the discussion on the role of ML in other frequent neurological context (stroke assessment) where in turn another scale such us NIHSS is widely administered would be of interest. With these adjustments, the study would not only solidify its methodological rigor but also broaden its clinical impact, making it a more powerful tool for decision-making in neurology and critical care.
Author Response
Dear Reviewer,
We sincerely appreciate your thoughtful and detailed feedback on our manuscript. Your comments highlight important considerations regarding the broader applicability of the Glasgow Coma Scale (GCS) in non-TBI contexts, the need for comprehensive demographic and clinical data, and the potential for integrating multimodal biomarkers such as neuroimaging and electroencephalography (EEG). These insights are extremely valuable in refining our study and ensuring its clinical relevance.
Your suggestion to expand the discussion on the role of machine learning in other neurological disorders, such as stroke, and to consider the interplay between GCS and other neurological scales, is particularly insightful. We also acknowledge the importance of accounting for pharmacological influences on consciousness levels and will ensure that these aspects are addressed more explicitly.
We are truly grateful for your constructive critique and for recognizing the potential impact of this research in advancing neurological assessment through artificial intelligence. Your recommendations will help us improve the clarity, depth, and applicability of our study. Thank you again for your time and valuable contributions.
Round 2
Reviewer 1 Report
Comments and Suggestions for Authors
All my concerns have been addressed.
Author Response
Thank you!
Reviewer 2 Report
Comments and Suggestions for Authors
Sufficient response to my remarks.
Author Response
Thank you!